# Seed Priming and Pericarp Removal Improve Germination in Low-Germinating Seed Lots of Industrial Hemp

**Jia W. Tan** [1] [iD], **Sharon T. Kester** [2], **Kai Su** [1], **David F. Hildebrand** [1] and **Robert L. Geneve** [2,*]

1. Department of Plant and Soil Science, University of Kentucky, Lexington, KY 405467, USA
2. Department of Horticulture, University of Kentucky, Lexington, KY 405467, USA
* Correspondence: rgeneve@uky.edu

**Abstract:** Industrial hemp (*Cannabis sativa* L.) is experiencing a resurgence in North America as an agricultural commodity. Germplasm improvement for locally adapted and stable cannabinoid production is an industry priority. This study used seed priming and pericarp removal to recover seedlings in low-germinating seed lots that could prove important for plant breeding and germplasm conservation. Both seed priming and pericarp removal improved early seed germination, but pericarp removal was more effective in improving overall final germination percentages. On average, pericarp removal improved final germination in low-germinating seed lots by approximately 38% compared to intact seed germination. In seeds with the pericarp removed, the initial germination substrate had an impact on normal seedling development following transplanting: those germinated for 2 to 3 days on rolled towels produced more normal seedlings compared to those started in Petri dishes. There was a dramatic increase in abnormal seedlings produced in the low-germinating seed lot initially germinated in a Petri dish wetted with 6 mL water where nearly 80% never transitioned to normal actively growing seedlings.

**Keywords:** seed vigor; osmotic priming; solid matrix priming; viability; germplasm





## 1. Introduction

Industrial hemp (*Cannabis sativa* L.) has been cultivated by humans as a source of fiber, food, oil, and medicines for thousands of years [1]. Despite the long history of cultivation, only recently has hemp cultivation resumed in North America [2]. Federal U.S. legislation has been passed that legalizes production of industrial hemp. The increasing demand for industrial hemp-related products requires breeders to develop regionally adapted varieties for specific end-use applications [3,4]. The hemp and related crop industry is one of the fastest-growing industries in the U.S.; therefore much attention has been directed to germplasm collection. There are several *Cannabis* collections currently available, such as the CPRO (now the Wageningen University) germplasm collection [5], the VIR collection in Europe [6], the INF&MP gene bank [7], and the YAAS collection [8]. Recently, Cornell University in Ithaca, New York has been designated the U.S. *Cannabis* germplasm repository. Additionally, due to the complexities associated with *Cannabis* germplasm regeneration, efficient methods of seed storage are also an important research area for the industry [9].

The dispersal unit in hemp is a multilayered pericarp surrounding an individual seed. Environmental conditions during seed production as well as storage conditions can impact seed quality as measured by viability and vigor. Elias et al. [10] observed a difference in seed quality in hemp seed harvested at different degrees of maturity related to position on the plant. Hemp shows orthodox seed behavior and can have extended storage life when stored at low temperature and low moisture content. However, it can be short-lived (less than 3 months) when stored at ambient temperature and moisture contents above 10% [11]. Similar to other crops, prolonged or non-optimal storage can lead to seed deterioration. Seed deterioration is a natural process involving cytological, physiological, biochemical,

and physical changes in seeds during aging [12]. These changes can give rise to reduced germinability and vigor loss [13–15]. During aging, seeds can become hypersensitive to adverse storage conditions, show slow and uneven germination, poor seedling emergence, and abnormal seedling growth. The ultimate result of seed deterioration is the inability to germinate [13,16].

Seed vigor represents those properties in a seed lot that impact rapid and uniform germination. These properties are lost during seed storage prior to a significant loss in germination. One method that appears to recover seed vigor in deteriorated seeds is seed priming [17]. Methods used to prime seeds include hydro-priming [18], osmotic priming [19,20], solid matrix priming [21], and halopriming [22].

Pericarp removal (dehulling) can be a mechanism to eliminate putative germination inhibitors or reduce the physical resistance of the seed coverings, allowing germination. Numerous studies have shown enhanced germination after pericarp removal, especially in grasses (Graminaceae) such as Japanese rice (*Oryza sativa*) [23], Indian ricegrass (*Oryzopsis hymenoldes*) [24], phalaris (*Phalaris tuberosa*) [25], and blue grama (*Bouteloua gracilis*) [26] as well as several members of the Asteraceae, such as zinnia (*Zinnia elegans*) [27] and sunflower (*Helianthus annuus*) [28]. Most of these studies involved species where pericarp removal improved germination by influencing innate seed dormancy rather than its influence on deteriorated, poor-germinating seed lots.

In this study, we compared seed priming and pericarp removal on the ability to increase seed vigor and seedling recovery in poor-germinating seed lots of industrial hemp assumed to be due to variable storage environments. Priming may recover germination by increasing embryo growth potential to permit radicle protrusion through the seed-covering layer(s), while pericarp removal may improve germination by reducing the physical restraint imposed by the seed coverings.

## 2. Materials and Methods

### 2.1. Plant Materials

Industrial hemp (*Cannabis sativa* L.) seeds (achenes) of Victoria, Finola, and Canda were supplied by Atalo, Winchester, KY, USA. All seeds were stored in sealed plastic bags at 10 °C.

### 2.2. Germination Conditions

Standard seed germination was in plastic Petri dishes (100 mm × 15 mm) containing 2 pieces of Grade 8001 germination paper (Stults Scientific Co., Springfield, IL, USA) moistened with deionized water and sealed with parafilm (Bemis Flexible Packaging, Shirley, MA, USA). Dishes were placed in a lighted incubator (8 h light, 16 h dark at approximately 60 µmol m$^{-2}$ s$^{-1}$) at 20–30 °C [29].

### 2.3. Tetrazolium Testing

Three replications of 25 randomly selected seeds were subjected to 2,3,5 triphenyl tetrazolium chloride (TZ) staining. Seeds were hydrated for 24 h prior to bisecting seeds longitudinally through the pericarp, seed coat, and embryo. One-half of each bisected seed was submerged in 0.1% 2,3,5-triphenyl tetrazolium chloride solution and incubated for 3 h at 35 °C. Evaluation of stained embryos followed the criteria of AOSA [30].

### 2.4. Seed Priming

Seeds were osmotically primed in a 250 mM (−1.15 mPa) aerated solution of NaCl for 4 days at 10 °C. Additionally, seeds were exposed to a solid matrix priming system utilizing Micro-Cel E (Celite Corp., Lompoc, CA, USA) at a ratio (by gram weight) of seeds to a carrier to double-distilled water of 4.0:3.2:5.6 (−1.16 mPa). Seeds were held for 3 days at 10 °C. Following priming, seeds were rinsed to remove priming materials and air-dried at room temperature under a laminar flow hood for 24 h. There were four replicate Petri dishes per treatment with 25 seeds per dish and experiments were repeated.

### 2.5. Pericarp Removal

Pericarp was removed using a standard bent nose plier (Stanley Tools) to crack open the pericarp to expose the seed from dry fruits. The exposed seed was removed from the fruit using forceps. The isolated seeds consisted of cotyledons and radicle with some of the papery seed coat attached. Twenty isolated seeds were placed in each Petri dish. There were 10 replicate dishes per treatment and the experiment was repeated. Germination in intact seeds was recorded after radicle protrusion, while a radicle length of 3 mm was considered germination in isolated embryos. Germination was recorded at 16, 40, 56, and 72 h and significance was determined using a single degree of freedom F-tests.

### 2.6. Initial Germination Substrate and Normal Seedling Development

Seeds from Victoria and Canda 2015 were sown in saturated rolled towels or Petri dishes wetted with 3 or 6 mL water and placed in germination incubators previously described. After 3 days, germinated individuals were measured for seedling length and 30 seedlings were transplanted to containers with greenhouse substrate (Pro-Mix; Premier Tech Horticulture, PA, USA) and moved to a growth chamber with 16 h light (approximately 150 $\mu$mol m$^{-2}$ s$^{-1}$) and 8 h dark cycles at 21 °C. After five days, they were evaluated for emergence and normal seedling development. Seedlings were considered normal if they transitioned into plants with multiple true leaves.

### 2.7. Statistics

For standard germination and priming experiments there were 25 seeds per Petri dish and 20 seeds per dish in the pericarp removal experiment. Significance was determined using single degree of freedom F-tests in SAS (SAS Institute, Cary, NC, USA) and mean separation was by Tukey's test at the 5% level. Percentages were transformed (arcsine of the square root of the proportion) where appropriate prior to statistical analysis using SigmaPlot 12.3 (Systat Software, Richmond, CA, USA).

## 3. Results

### 3.1. Germination and Tetrazolium Staining

Only Victoria showed high viability based on standard germination tests and TZ staining (Tables 1 and 2). The other hemp seed lots used in this study germinated at less than 25% under standard germination conditions and had only a small percentage (~15%) that stained darkly during the TZ test. A large proportion of embryos in these seed lots were lightly stained during TZ testing, suggesting they had low vigor and were losing viability.

**Table 1.** The degree of tetrazolium (TZ) staining in seeds from four industrial hemp cultivars.

| | Hemp Seed Line | | | |
|---|---|---|---|---|
| TZ Staining (%) | Victoria | Finola | Canda 2014 | Canda 2015 |
| Fully stained [z] | 76 | 16 | 8 | 15 |
| Variable | 24 | 72 | 60 | 56 |
| Unstained | 0 | 12 | 32 | 29 |

[z] Fully stained embryos showed uniform dark staining; variable stained embryos showed partial dark staining or overall light staining; unstained embryos did not show any color.

**Table 2.** Germination percentage over time in industrial hemp seeds following osmotic priming.

| | Victoria | | Finola | | Canda 2014 | | Canda 2015 | |
|---|---|---|---|---|---|---|---|---|
| Time (h) | Untreated | Primed | Untreated | Primed | Untreated | Primed | Untreated | Primed |
| 16 | 0 | 49 * | 0 | 18 * | 0 | 7 * | 0 | 15 * |
| 40 | 71 | 82 * | 8 | 20 * | 5 | 11 * | 18 | 26 * |
| 56 | 76 | 90 * | 9 | 22 * | 9 | 12 | 19 | 26 * |
| 72 | 86 | 92 | 13 | 24 | 10 | 12 | 24 | 26 |

* Indicates differences between paired means for untreated and primed seeds for each seed line and time by single degree of freedom F-tests ($p \leq 0.05$).

### 3.2. Seed Priming

Regardless of the priming method, primed seed lots initiated germination prior to untreated seeds (Tables 2 and 3). At 16 hours, all primed seed lots showed significantly increased germination percentages compared to untreated seeds. The same significant increase in germination percentage across all four seed lots compared to untreated seeds was observed after 40 and 56 h except in osmotically primed Canda 2014 at 56 h and solid matrix primed Finola at 40 and 56 h. For osmotically primed seeds, there was a trend in all seed lots for increased final germination percentages, but the data were not significant (Table 2). There was also a trend for increased final germination in solid matrix-primed seed lots, but there was only a significant increase in final germination percentages for Canda 2014 and Canda 2015 (Table 3).

**Table 3.** Germination percentage over time in industrial hemp seeds following solid matrix priming.

| Time (h) | Victoria | | Finola | | Canda 2014 | | Canda 2015 | |
|---|---|---|---|---|---|---|---|---|
| | Untreated | Primed | Untreated | Primed | Untreated | Primed | Untreated | Primed |
| 16 | 1 | 26 * | 1 | 6 * | 1 | 6 * | 1 | 5 * |
| 40 | 64 | 85 * | 16 | 23 | 4 | 23 * | 9 | 25 * |
| 56 | 79 | 90 * | 17 | 25 | 6 | 25 * | 11 | 31 * |
| 72 | 87 | 90 | 17 | 25 | 6 | 25 * | 12 | 31 * |

* Indicates differences between paired means for untreated and primed seeds for each seed line and time by single degree of freedom F-tests ($p \leq 0.05$).

### 3.3. Pericarp Removal

Simple pericarp removal dramatically increased early germination in all seed lots (Table 4). Except for Victoria, the other seed lots showed higher final germination percentages after pericarp removal compared to untreated seeds. Collectively, this resulted in approximately a 38% increase in final germination for Finola, Canda 2014, and Canda 2015 compared to intact seed germination. If it is assumed that all TZ-stained embryos regardless of staining intensity had the capacity to germinate, then pericarp removal recovered 32%, 31%, and 69% of germination potential for Finola, Canda 2014, and Canda 2015, respectively.

**Table 4.** Germination percentage over time in industrial hemp seeds following pericarp removal.

| Time (h) | Victoria | | Finola | | Canda 2014 | | Canda 2015 | |
|---|---|---|---|---|---|---|---|---|
| | Untreated | No Pericarp | Untreated | No Pericarp | Untreated | No Pericarp | Untreated | No Pericarp |
| 16 | 0.5 | 76 * | 2 | 9 | 0.5 | 6 * | 2 | 29 * |
| 40 | 77 | 90 * | 14 | 21 * | 8 | 16 | 23 | 40 * |
| 56 | 87 | 93 | 16 | 26 * | 11 | 20 * | 27 | 47 * |
| 72 | 93 | 96 | 18 | 28 * | 13 | 21 * | 29 | 49 * |

* Indicates differences between paired means for untreated and primed seeds for each seed line and time by single degree of freedom F-tests ($p \leq 0.05$).

In the high-germinating Victoria seed lot, initial seedling growth prior to transplanting, seedling emergence after transplanting, and reduced abnormal seedling development was seen in those individuals initially started on rolled towels compared to those started on Petri dishes (Table 5). The same trend was seen in the low-germinating Canda seed lot, with the poorest seedling emergence and highest number of abnormal seedlings observed in individuals started in Petri dishes (Table 5).

**Table 5.** Victoria and Canda 2015 hemp seedling emergence percentage and initial seedling length on rolled towels or in Petri dishes with 3 or 6 mL of water before being transplanted to a greenhouse substrate.

| Cultivar | Treatment | Substrate | Seedling Length (cm) | Seedling Emergence (%) | Abnormal (%) |
|---|---|---|---|---|---|
| Victoria | Intact | Towel | 2.37 [ab,*] | 86.7 [ab] | 0 |
| | No pericarp | Towel | 2.88 [a] | 96.7 [a] | 0 |
| | Intact | Dish 3 mL | 2.00 [b] | 96.7 [a] | 0 |
| | No pericarp | Dish 3 mL | 1.51 [b] | 86.7 [ab] | 10.3 |
| | Intact | Dish 6 mL | 1.76 [b] | 80.0 [b] | 15.4 |
| | No pericarp | Dish 6 mL | 1.62 [b] | 66.7 [c] | 35.0 |
| Canda | Intact | Towel | 1.86 [b] | 33.3 [ab] | 10.0 |
| | No pericarp | Towel | 2.56 [a] | 43.3 [a] | 20.0 |
| | Intact | Dish 3 mL | 1.12 [b] | 33.3 [ab] | 25.0 |
| | No pericarp | Dish 3 mL | 1.65 [b] | 36.7 [ab] | 23.1 |
| | Intact | Dish 6 mL | 1.56 [b] | 26.7 [b] | 45.5 |
| | No pericarp | Dish 6 mL | 1.02 [b] | 20.0 [b] | 83.3 |

* Means within a column for each seed line followed by the same letter were not different at 5% level by Tukey's test.

## 4. Discussion

### 4.1. Germination

Germination starts with the uptake of water by the dry seed and typically concludes with radicle protrusion through the seed coverings (endosperm and seed coat/pericarp) [31,32]. Cell elongation is necessary and generally accepted to be sufficient for the completion of radicle protrusion (visible germination) [33,34]. Endosperm and the seed coverings (seed coat/pericarp) act as physical barriers to seed germination. A decline in physical resistance in the micropylar endosperm appears to be a prerequisite for radicle protrusion in many seeds [34–37]. Hence, an increase in embryo growth potential or weakening of the physical constraint associated with the seed-covering layer can accelerate germination and/or increase final germination [33,37–39].

### 4.2. Seed Priming

Seed priming may impact both growth potential and seed covering strength, but it is thought that increased growth potential is the major mechanism responsible for the priming effect. In this study, using low-germinating seed lots, priming reduced the time to radicle protrusion and there was a trend for increased final germination, especially using solid matrix priming (Table 3). Increased time to radicle protrusion has been suggested as an indication of declining seed vigor and a prelude to loss in seed viability [40]. Therefore, priming-induced recovery of seed vigor may be responsible for the trend in improved germination observed in this study [17]. Solid matrix priming was more effective in improving final germination compared to osmotic priming (Tables 2 and 3). Micro-Cel E used for solid matrix priming is a calcium silicate and the available calcium may be accountable for the differences seen between the two priming methods [41]. Calcium is essential for regulation of many physiological and cellular events and calcium applications have significantly influenced seed germination [42,43].

### 4.3. Pericarp Removal

Pericarp removal increased both germination speed and overall germination percentage (Table 4). Compared to the two priming methods, pericarp removal was more effective in enhancing final germination in the low-germinating seed lots. The results suggest that physical restraint imposed by the seed coverings had a major impact on limiting germination in poor-germinating seed lots. Once the pericarp was removed from hemp seeds, the remaining seed coverings were not a significant barrier to germination and the embryo/seedling was observed to break out of the true seed coat by expansion of the

hypocotyl and cotyledons rather than the usual radicle protrusion seen in intact seeds with pericarp coverings intact (Figure 1). Taken together, this suggests that removal of the pericarp was more effective than priming for increasing final germination percentages.

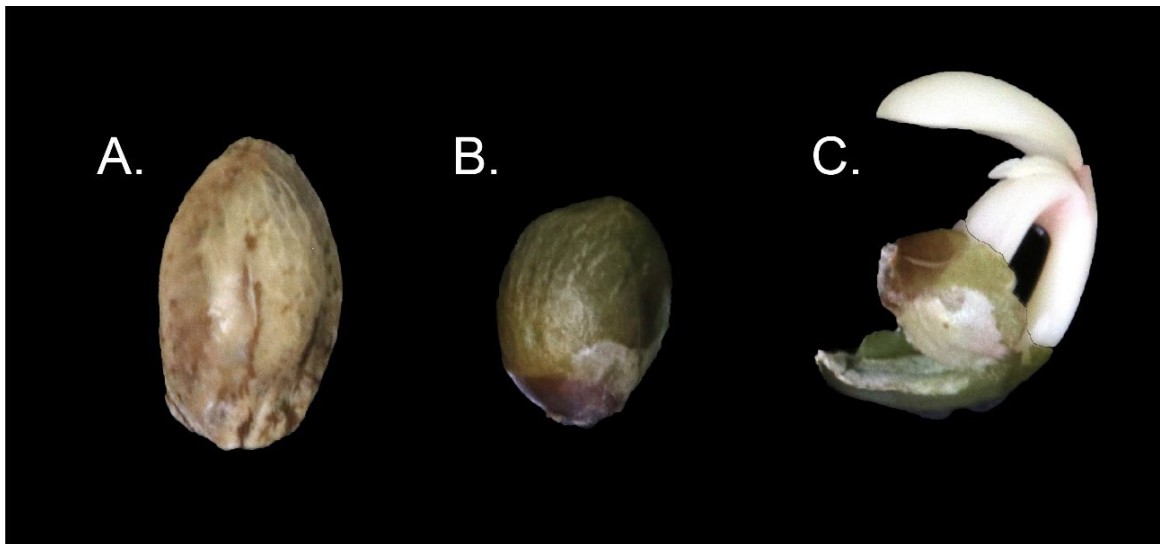

**Figure 1.** Germination in hemp after pericarp removal. (**A**) Intact diaspore with pericarp intact. (**B**) True seed after pericarp removal. (**C**) Germination in a seed after pericarp removal showing the expansion of the hypocotyl and cotyledons pushing the seedling from the true seed.

One issue with pericarp removal to recover germination has been the impact on normal seedling development. Pericarp removal in spinach was not as effective as hydrogen peroxide treatment for increasing germination percentages primarily because pericarp removal led to more abnormal seedlings [44]. In the current study, there was an interaction between pericarp removal and initial germination substrate moisture level on subsequent normal seedling development (Table 5). This was more evident in the low-germination Canda seed lot, where 80% of the transplanted seedlings were abnormal in seeds with pericarp removed initially germinated in Petri dishes with 6 mL water, possibly due to lower initial oxygen availability.

## 5. Conclusions

Germplasm improvement for locally adapted and stable cannabinoid production is an industry priority. Loss of germination capacity limits the ability to conserve important genetics preserved in the seed. Therefore, there is the potential need for recovery of unique germplasm for hemp seed lots that have diminished viability due to storage under less-than-ideal conditions. In the current study, pericarp removal was effective at recovering normal seedlings in low-germinating seed lots. Using this technique, our research group was able to recover a few seedlings in a genetic resource with a limited seed supply that would not germinate under standard germination conditions. Pericarp removal provides another tool for germplasm collections and breeding programs to recover valuable genetics in deteriorating seed lots.

**Author Contributions:** R.L.G. conceived and supervised the project. J.W.T., K.S., S.T.K. performed the research and evaluated data. J.W.T. and R.L.G. wrote the article; D.F.H. revised the article. All authors have read and agreed to the published version of the manuscript.

**Funding:** This research received no external funding.

**Data Availability Statement:** The data presented in this study are available on request from the corresponding author.

**Acknowledgments:** This project was supported by the USDA National Institute of Food and Agriculture, hatch project number KY011042.

**Conflicts of Interest:** The authors declare no conflict of interest.

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
