# Peer review of "Seed Priming and Pericarp Removal Improve Germination in Low-Germinating Seed Lots of Industrial Hemp"

_2673-7655, doi:10.3390/crops2040028_

Round 1

Reviewer 1 Report

This manuscript provides some data that can be useful for managing hemp germplasm, but some important aspects must be improved.

Firstly, although throughout the manuscript, starting from the title, ‘pericarp removal’ is mentioned as a technique used to improve germination, in subsection 2.5 the method actually applied under the name ‘pericarp removal’ is described as: “Pericarp was removed using a standard bent nose plier (Stanley Tools) to crack open the pericarp to expose the seed. The exposed embryo was removed from the fruit using forceps. The isolated embryo consisted of cotyledons and radicle with some of the papery seed coat attached. Twenty isolated embryos or pericarp-intact seeds were placed in each petri dish” (lines 96-100). This description corresponds to the well-known technique of “embryo rescue”, which is so old and widespread that even Wikipedia has a page for it (https://en.wikipedia.org/wiki/Embryo_rescue) where it is explained that “Embryo rescue is one of the earliest and successful forms of in-vitro culture techniques that is used to assist in the development of plant embryos that might not survive to become viable plants” and “Embryo rescue was first documented in the 18th century”. So, it would seem that the present authors are trying to sell some old-known general fact as a novelty for hemp. Moreover, ‘pericarp removal’ is correctly illustrated in Figure 1B, where, indeed, no mention is made that “The exposed embryo was removed” after pericarp removal. The authors must clarify this key point and amend the incongruence between the method description and Figure 1.

For peeling the hard pericarp off, pre-moistening for one hour in distilled water has been used (Jacob et al. 2022. J. Natur. Fibers 19:1281-1286): did the present authors find similar problems or was moistening not necessary in the present case?

In some instances, throughout the manuscript, ‘Petri dish’ is written without the first letter capitalized (that is, “petri dish”): it should be changed to ‘Petri dish’ as ‘Petri’ is the name of the guy who invented this kind of vessel.

‘Seed vigor’ is mentioned throughout the manuscript, but no data are shown for it. Although end-of-test germination percentages are linked to vigour, the two are not the same thing (Reed et al. 2022. Heredity https://doi.org/10.1038/s41437-022-00497-2).

Line 41: change “The seed unit …” to “The dispersal unit …” or alike.

Seeds were germinated “at 20-30°C” (line 78): this is a quite wide temperature range, because hemp germination rate is different at 20 °C , 25 °C and 30 °C (Lisson et al. 2000. Austral. J. Exp. Agric. 40:405-411; Alipoor and Mahmodi 2015. Iran. J. Seed Res. 2:37-50; Barooti et al. 2018. Iran. J. Seed Sci. Technol. 7:127-136).

Statistical significance was “determined using a single degree of freedom F-tests in SAS” (line 104): this is equivalent to a t-test. Please, note that when comparing several means, a multiple comparison procedure is required. A t-test, in fact, does not control the family-wise error rate at the specified alpha level. It has, therefore, more power than other mean comparison tests (that is, it reduces the risk of failing to detect a non-negligible treatment effect), but only because it has a high risk of familywise Type I error, which means that the risk of falsely concluding that a treatment effect exists is too high (it controls the Type I error much less than Tukey's test). Kramer et al. (2016, J. Am. Soc. Hort. Sci. 141:400–406) advocated against such approach to ensure the soundness of a scientific paper. In the experimental conditions described in the manuscript, I’d typically recommend the Tukey's test.

Besides, statistical analysis should be a sub-section of the methods, as it was applied to every test, and more details must be provided (an ANOVA is required). Moreover, using these tests when dealing with percentages, particularly, germination percentages, sometimes requires that the angular (arcsin) transformation or more complex statistical approaches are used (Onofri et al. 2010. Current statistical issues in Weed Research. Weed Res. 50:5-24; Gianinetti 2020. Basic features of the analysis of germination data with generalized linear mixed models. Data 5:6). Further care must be given to the fact that several measures were done on the same subjects: repeated measures ANOVA should be used for longitudinal data (generalized linear mixed models, or survival analysis, would be even better).

Table 4 is partially outside of the page. Please, resize.

Line 183, “Endosperm and the seed coat (pericarp) act as physical barriers to seed germination”: please, note that, botanically, the seed coat is different from the pericarp.

Line 234, “genetics preserved in the seed”: please, better define this concept.

Lines 234-239: these are hardly conclusions; they are, in fact, important pieces of information that must be provided in the Introduction section to introduce the rationale of this study.

Author Response

This manuscript provides some data that can be useful for managing hemp germplasm, but some important aspects must be improved.

Firstly, although throughout the manuscript, starting from the title, ‘pericarp removal’ is mentioned as a technique used to improve germination, in subsection 2.5 the method actually applied under the name ‘pericarp removal’ is described as: “Pericarp was removed using a standard bent nose plier (Stanley Tools) to crack open the pericarp to expose the seed. The exposed embryo was removed from the fruit using forceps. The isolated embryo consisted of cotyledons and radicle with some of the papery seed coat attached. Twenty isolated embryos or pericarp-intact seeds were placed in each petri dish” (lines 96-100). This description corresponds to the well-known technique of “embryo rescue”, which is so old and widespread that even Wikipedia has a page for it (https://en.wikipedia.org/wiki/Embryo_rescue) where it is explained that “Embryo rescue is one of the earliest and successful forms of in-vitro culture techniques that is used to assist in the development of plant embryos that might not survive to become viable plants” and “Embryo rescue was first documented in the 18th century”. So, it would seem that the present authors are trying to sell some old-known general fact as a novelty for hemp. Moreover, ‘pericarp removal’ is correctly illustrated in Figure 1B, where, indeed, no mention is made that “The exposed embryo was removed” after pericarp removal. The authors must clarify this key point and amend the incongruence between the method description and Figure 1.

Thank you for this observation. Our intention was to isolate the seed from the fruit. Because the botanical seed coat is papery and can remain attached to the endocarp, we were generally removing seeds that could have some portion of the seed coat removed.

This section was edited “Pericarp was removed using a standard bent nose plier (Stanley Tools) to crack open the pericarp to expose the seed. The exposed seed was removed from the fruit using forceps. The isolated seed consisted of cotyledons and radicle with some of the papery seed coat attached. Twenty isolated seeds were placed in each petri dish.”

For peeling the hard pericarp off, pre-moistening for one hour in distilled water has been used (Jacob et al. 2022. J. Natur. Fibers 19:1281-1286): did the present authors find similar problems or was moistening not necessary in the present case?

Although pericarp was tedious, be were able to remove seed from dry fruits. We felt this was a better option especially when comparing germination over time that would have been impacted by pre-soaking.

In some instances, throughout the manuscript, ‘Petri dish’ is written without the first letter capitalized (that is, “petri dish”): it should be changed to ‘Petri dish’ as ‘Petri’ is the name of the guy who invented this kind of vessel.

My experience is that capitalization of Petri dish varies with the journal preference. I personally agree that it should be Petri dish and to be consistent, we have made those changes.

‘Seed vigor’ is mentioned throughout the manuscript, but no data are shown for it. Although end-of-test germination percentages are linked to vigour, the two are not the same thing (Reed et al. 2022. Heredity https://doi.org/10.1038/s41437-022-00497-2).

We agree that seed vigor is not the same as final germination. We are suggesting that priming is impacting vigor as indicated in pair-wise comparisons at each time point between primed and untreated seeds (Table 3) and this vigor resulted in higher final germination. I think this is an appropriate way to view the data in Table 3 and I believe we only use the term vigor in association with seed priming.

Line 41: change “The seed unit …” to “The dispersal unit …” or alike.

            This change has been made.

Seeds were germinated “at 20-30°C” (line 78): this is a quite wide temperature range, because hemp germination rate is different at 20 °C , 25 °C and 30 °C (Lisson et al. 2000. Austral. J. Exp. Agric. 40:405-411; Alipoor and Mahmodi 2015. Iran. J. Seed Res. 2:37-50; Barooti et al. 2018. Iran. J. Seed Sci. Technol. 7:127-136).

The germination was selected because it is the standard germination temperature indicated by ISTA. We have included information from the Alipoor reference to provide more information for the reader.

Statistical significance was “determined using a single degree of freedom F-tests in SAS” (line 104): this is equivalent to a t-test. Please, note that when comparing several means, a multiple comparison procedure is required. A t-test, in fact, does not control the family-wise error rate at the specified alpha level. It has, therefore, more power than other mean comparison tests (that is, it reduces the risk of failing to detect a non-negligible treatment effect), but only because it has a high risk of familywise Type I error, which means that the risk of falsely concluding that a treatment effect exists is too high (it controls the Type I error much less than Tukey's test). Kramer et al. (2016, J. Am. Soc. Hort. Sci. 141:400–406) advocated against such approach to ensure the soundness of a scientific paper. In the experimental conditions described in the manuscript, I’d typically recommend the Tukey's test.

Besides, statistical analysis should be a sub-section of the methods, as it was applied to every test, and more details must be provided (an ANOVA is required). Moreover, using these tests when dealing with percentages, particularly, germination percentages, sometimes requires that the angular (arcsin) transformation or more complex statistical approaches are used (Onofri et al. 2010. Current statistical issues in Weed Research. Weed Res. 50:5-24; Gianinetti 2020. Basic features of the analysis of germination data with generalized linear mixed models. Data 5:6). Further care must be given to the fact that several measures were done on the same subjects: repeated measures ANOVA should be used for longitudinal data (generalized linear mixed models, or survival analysis, would be even better).

We very much agree with the reviewer that Tukey’s is an appropriate and conservative mean separation statistic when comparing multiple comparisons. This is why we chose Tukey’s test for the data reported in Table 5. However, in the pericarp removal and priming experiments, we were not comparing means across seed lots over time. We were doing a single pair-wise comparison between treatments within the same seed lot at the same time point. Since there are only two data points the ANOVA comparison is a single-degree of freedom test for significance. We feel this is the appropriate statistic but we could report a T-test or Tukey statistic if that is necessary.

A statistics section was included.

Table 4 is partially outside of the page. Please, resize.

Done.

Line 183, “Endosperm and the seed coat (pericarp) act as physical barriers to seed germination”: please, note that, botanically, the seed coat is different from the pericarp.

The line was edited for clarity. “Endosperm and seed coverings (seed coat / pericarp) act as physical barriers to seed germination”.

Line 234, “genetics preserved in the seed”: please, better define this concept.

Additional information was provided for clarity.

Lines 234-239: these are hardly conclusions; they are, in fact, important pieces of information that must be provided in the Introduction section to introduce the rationale of this study.

Information was moved to the introduction.

Reviewer 2 Report

The germination process consists of several stages, but the authors limited themselves to the physiological stage that ends with the formation of a sprout. It would be interesting to investigate the physical (swelling) and biochemical stages in which the reserve substances leading to the formation of the sprout are activated.

The title of the work is consistent with the article presented for evaluation.

The experiment was planned correctly, and the authors described the research methods in a concise and clear manner. The empirical part and the discussion of the results were constructed in accordance with the applicable scheme and properly conducted.

There are, however, a few questions and comments to the Authors:

1. Why were these cannabis strains selected for the research? How long were the seeds stored at Atalo, Winchester KY? Was germination also determined immediately after harvest?

2. There is no precise presentation of the relationship between germination capacity and speed as a function of time. Only 4 points were set. Why was data only collected between 14 and 18 hours?

3. The authors report that seeds of the Canda 2015 and Viktoria variety were sown on rolled towels and Petri dishes. Why was Finola and Canda 2014 not mentioned? Why were they wetted with different volumes of water (3 or 6 ml)?

4. How many times was the experiment repeated?

5. The cited literature is not up-to-date, out of 43 it includes only 2 items from the 5-year period 2017-2022. Should be updated.

6. IF the author of the photo from Figure 1 is not a co-author of the work, his / her data should be provided.

Author Response

The germination process consists of several stages, but the authors limited themselves to the physiological stage that ends with the formation of a sprout. It would be interesting to investigate the physical (swelling) and biochemical stages in which the reserve substances leading to the formation of the sprout are activated.

It would be interesting to study additional aspects of hemp germination, but this was not the objective of the current study.

The title of the work is consistent with the article presented for evaluation.

The experiment was planned correctly, and the authors described the research methods in a concise and clear manner. The empirical part and the discussion of the results were constructed in accordance with the applicable scheme and properly conducted.

There are, however, a few questions and comments to the Authors:

  1. Why were these cannabis strains selected for the research? How long were the seeds stored at Atalo, Winchester KY? Was germination also determined immediately after harvest?

These were the available cannabis strains available at the time and their seed quality was variable. The seed company could not provide the original seed germination of these lots.

  1. There is no precise presentation of the relationship between germination capacity and speed as a function of time. Only 4 points were set. Why was data only collected between 14 and 18 hours?

The initial starting point for evaluating germination was 16 hours because germination in untreated seeds was generally around 20 hours. We agree that additional information would be provided if we included additional data for 14 and 18 hours, but our main objective was to investigate ultimate germination percentage recovery in low germinating seed lots by priming rather than their early germination.

  1. The authors report that seeds of the Canda 2015 and Viktoria variety were sown on rolled towels and Petri dishes. Why was Finola and Canda 2014 not mentioned? Why were they wetted with different volumes of water (3 or 6 ml)?

The main reason for including only Canda 2015 and Victroria was available seeds in the provided seed lots. We also felt that Canda 2015 was representative of a low germinating seed lot based on our data.

Two levels of water in the Petri dishes was used because in preliminary studies we noticed that excess water in the dishes may have been reducing seedling morphology possible because of lower availability of oxygen. A statement was added on line 223.

  1. How many times was the experiment repeated?

Pericarp removal and priming experiments were repeated as indicated in the Materials and Methods.

  1. The cited literature is not up-to-date, out of 43 it includes only 2 items from the 5-year period 2017-2022. Should be updated.

            Additional literature was included in the introduction.

  1. IF the author of the photo from Figure 1 is not a co-author of the work, his / her data should be provided.

The image is the authors.

Round 2

Reviewer 1 Report

As the aim of this study was to find a method “to recover seedlings in low-germinating seed lots” (line 12) and “seeds were stored in sealed plastic bags at 10°C” (lines 81-82), it would be useful if the authors can provide more details about the reason the seed batches used in this study (actually, three out of four) were low-germinating. On the one hand, the authors mention several published studies “where pericarp removal improved germination by influencing innate seed dormancy rather than” because of "its influence on deteriorated ... seed lots” (lines 69-71). So, did the authors ascertain the presence of dormancy in their hemp seeds? On the other hand, the authors clarify that “Hemp ... seed … can have extended storage life when stored at low temperature and low moisture content. However, it can be short-lived (less than 3 months) when stored at ambient temperature and moisture contents above 10%” (lines 45-48). Thus, it appears the authors are suggesting that these seed batches had been stored at high temperature and/or moisture content before the seeds were made available to the authors. Even though it is probably difficult to recover details about prior storage conditions, I recommend the authors clearly state what the expected cause for which these seed batches were low-germinating was. In fact, hemp seeds can have germination problems, but this is commonly due to a high presence of immature seeds. To clearly define what the problem is, is the first step when one wishes to propose a solution.

Lines 81-82, "All seeds were stored in sealed plastic bags at 10°C": at what MC?

I still find quite perplexing the description, in subsection 2.5, of the method actually used to improve germination and described under the name ‘pericarp removal’. Specifically, “The exposed seed was removed from the fruit using forceps. The isolated seeds consisted of cotyledons and radicle with some of the papery seed coat attached” (lines 106-108): I miss the part wherein most seed coat has been removed leaving only “some of the papery seed coat attached”. Once the pericarp has been carefully removed, the seed with the whole seed coat should be obtained (as shown in Figure 1B). In their response, the authors state that “Our intention was to isolate the seed from the fruit. Because the botanical seed coat is papery and can remain attached to the endocarp, we were generally removing seeds that could have some portion of the seed coat removed”. This is OK, but is must be clarified in the manuscript. Furthermore, if most seed coat had been removed, then the embryo (as stated in the previous version) was actually used for germination tests, and this would be practically equivalent to the well-known technique of “embryo rescue”, as I previously mentioned. So, please, clearly describe in the manuscript what has been done. If most seed coat had been removed, please, mention that this technique is similar to the “embryo rescue” and quote some basic reference on embryo rescue”.

Table 1: please note that for “Canda 2015” the percentages sum up to 101%.

Seeds were germinated “at 20-30°C” (line 87): as remarked in my previous review, this is a quite wide temperature range, because hemp germination rate is different at 20 °C , 25 °C and 30 °C (Lisson et al. 2000. Austral. J. Exp. Agric. 40:405-411; Alipoor and Mahmodi 2015. Iran. J. Seed Res. 2:37-50; Barooti et al. 2018. Iran. J. Seed Sci. Technol. 7:127-136). Even if this “is the standard germination temperature indicated by ISTA” (as stated in the authors’ response), these differences in temperature could be large enough to produce significantly different germination values unless all the treatments had been done at the same time. So, please, clarify this aspect.

It is not clear how many seeds and replicate dishes have been used for data shown in Table 2. This holds true for data shown in Table 2. Please, provide details.

Statistical significance was “determined using a single degree of freedom F-tests” (line 113): this is equivalent to a t-test. As several measures were done on the same subjects, repeated measures ANOVA or other specific statistical procedure for longitudinal data should be applied rather than a single degree of freedom F-test or the equivalent t-test (Onofri et al. 2010. Current statistical issues in Weed Research. Weed Res. 50:5-24). In the present case, as the results are quite clear, I am not going to insist further on this aspect, but, no, it is neither true that “we were not comparing means across seed lots over time” (because you were) nor that a single-degree of freedom test for significance “is the appropriate statistic” (because it is not; although it is not terribly wrong too), as stated in the authors’ response.

Lines 128-129, “Percentages were transformed (square of the arc-sine) where …”: please, note that the angular transformation is calculated from the arcsine of the square root of the proportion (Sokal and Rohlf 1969. Biometry: The Principles and Practice of Statistics in Biological Research), not the “square of the arc-sine”.

Line 191: please, note that “Seedling emergence” rather than “seed germination percentage” was assessed here.

Table 5: “Days after sowing” does not seem to be correct here.

Line 259, “recovery of unique germplasm for hemp seed lots that have lost viability”: once viability is lost, no method can resurrect the seeds.

Author Response

As the aim of this study was to find a method “to recover seedlings in low-germinating seed lots” (line 12) and “seeds were stored in sealed plastic bags at 10°C” (lines 81-82), it would be useful if the authors can provide more details about the reason the seed batches used in this study (actually, three out of four) were low-germinating. On the one hand, the authors mention several published studies “where pericarp removal improved germination by influencing innate seed dormancy rather than” because of "its influence on deteriorated ... seed lots” (lines 69-71). So, did the authors ascertain the presence of dormancy in their hemp seeds? On the other hand, the authors clarify that “Hemp ... seed … can have extended storage life when stored at low temperature and low moisture content. However, it can be short-lived (less than 3 months) when stored at ambient temperature and moisture contents above 10%” (lines 45-48). Thus, it appears the authors are suggesting that these seed batches had been stored at high temperature and/or moisture content before the seeds were made available to the authors. Even though it is probably difficult to recover details about prior storage conditions, I recommend the authors clearly state what the expected cause for which these seed batches were low-germinating was. In fact, hemp seeds can have germination problems, but this is commonly due to a high presence of immature seeds. To clearly define what the problem is, is the first step when one wishes to propose a solution.

These seed lots were obtained from a commercial source that were attempting to increase their germplasm collection. My understanding was that they were obtained second hand from other seed producers. It is not possible to obtain information on their storage environment. Based on the TZ reported in Table 1, it is clear that seed lots were deteriorating. We can assume this was due to poor storage environments, but since we did not have initial germination and viability following harvest this is only an assumption. We added a sentence on line 71 indicating this assumption.

Lines 81-82, "All seeds were stored in sealed plastic bags at 10°C": at what MC?

All our seeds are stored at less than 5% moisture content. However, we did not specifically measure moisture content in these seeds partially because we did not control their storage for years prior to obtaining them for our studies.

I still find quite perplexing the description, in subsection 2.5, of the method actually used to improve germination and described under the name ‘pericarp removal’. Specifically, “The exposed seed was removed from the fruit using forceps. The isolated seeds consisted of cotyledons and radicle with some of the papery seed coat attached” (lines 106-108): I miss the part wherein most seed coat has been removed leaving only “some of the papery seed coat attached”. Once the pericarp has been carefully removed, the seed with the whole seed coat should be obtained (as shown in Figure 1B). In their response, the authors state that “Our intention was to isolate the seed from the fruit. Because the botanical seed coat is papery and can remain attached to the endocarp, we were generally removing seeds that could have some portion of the seed coat removed”. This is OK, but is must be clarified in the manuscript. Furthermore, if most seed coat had been removed, then the embryo (as stated in the previous version) was actually used for germination tests, and this would be practically equivalent to the well-known technique of “embryo rescue”, as I previously mentioned. So, please, clearly describe in the manuscript what has been done. If most seed coat had been removed, please, mention that this technique is similar to the “embryo rescue” and quote some basic reference on “embryo rescue”.

I have worked with a number of species with pericarps as part of the seed unit. When removing the pericarp from dry fruits it is usual that some of the seed coat remains attached to the pericarp wall. We have added information on line 103 to indicate removal was from dry seed to make the removal process more specific. I would prefer not using the term embryo rescue for this type of technique. It is true that in the older literature, embryo removal from dormant seeds, especially those with deep physiological dormancy was referred to as embryo rescue. However, in recent nomenclature, embryo rescue refers to the in vitro removal of immature embryos from wide genetic crosses that would not reach viable maturity. I did a quick Google Scholar search and all papers referred to embryo rescue as this process.

Table 1: please note that for “Canda 2015” the percentages sum up to 101%.

Thankyou, it has been changed.

Seeds were germinated “at 20-30°C” (line 87): as remarked in my previous review, this is a quite wide temperature range, because hemp germination rate is different at 20 °C , 25 °C and 30 °C (Lisson et al. 2000. Austral. J. Exp. Agric. 40:405-411; Alipoor and Mahmodi 2015. Iran. J. Seed Res. 2:37-50; Barooti et al. 2018. Iran. J. Seed Sci. Technol. 7:127-136). Even if this “is the standard germination temperature indicated by ISTA” (as stated in the authors’ response), these differences in temperature could be large enough to produce significantly different germination values unless all the treatments had been done at the same time. So, please, clarify this aspect.

We recognize this but with limited seeds available ISTA recommendations seemed appropriate temperature for this study and has been used by other researchers in hemp (Elias, S.G.; Wu, Y.; Stimpson, D. Seed quality and dormancy of hemp (Cannabis staiva L.). J. Agric. Hemp Research, 2020, 2, 1-15). The differences due to temperature do not become significant until after 30C as indicated by our sister publication under review.

It is not clear how many seeds and replicate dishes have been used for data shown in Table 2. This holds true for data shown in Table 2. Please, provide details.

Replication is indicated in the Materials and Methods lines 98,99

Statistical significance was “determined using a single degree of freedom F-tests” (line 113): this is equivalent to a t-test. As several measures were done on the same subjects, repeated measures ANOVA or other specific statistical procedure for longitudinal data should be applied rather than a single degree of freedom F-test or the equivalent t-test (Onofri et al. 2010. Current statistical issues in Weed Research. Weed Res. 50:5-24). In the present case, as the results are quite clear, I am not going to insist further on this aspect, but, no, it is neither true that “we were not comparing means across seed lots over time” (because you were) nor that a single-degree of freedom test for significance “is the appropriate statistic” (because it is not; although it is not terribly wrong too), as stated in the authors’ response.

Acknowledged

Lines 128-129, “Percentages were transformed (square of the arc-sine) where …”: please, note that the angular transformation is calculated from the arcsine of the square root of the proportion (Sokal and Rohlf 1969. Biometry: The Principles and Practice of Statistics in Biological Research), not the “square of the arc-sine”.

Done

Line 191: please, note that “Seedling emergence” rather than “seed germination percentage” was assessed here.

Done

Table 5: “Days after sowing” does not seem to be correct here.

Thank you, it was removed

Line 259, “recovery of unique germplasm for hemp seed lots that have lost viability”: once viability is lost, no method can resurrect the seeds.

Changed lost to diminished on line 253

Reviewer 2 Report

The Authors addressed the comments of the reviewer and the responses are partially satisfactory. However, the Authors added only one item in References from 2017-2022. This still needs to be expanded.

Author Response

The Authors addressed the comments of the reviewer and the responses are partially satisfactory. However, the Authors added only one item in References from 2017-2022. This still needs to be expanded.

We felt that the additional references added to the paper were the most appropriate. If the reviewer has specific references they would like to see included, please let us know. There are already 45 references for this brief manuscript and I was wondering if we were providing too many references.